# Pancytopenia Secondary to Vitamin B12 Deficiency in Older Subjects

**DOI:** 10.3390/jcm12052059

**Published:** 2023-03-06

**Authors:** Giulia Costanzo, Giada Sambugaro, Giulia Mandis, Sofia Vassallo, Angelo Scuteri

**Affiliations:** 1S.C. di Medicina Interna, Policlinico Universitario Monserrato “Duilio Casula”—AOU di Cagliari, 09123 Cagliari, Italy; 2Scuola Specializzazione Allergologia e Immunologia Clinica, Universita’ di Cagliari, 09124 Cagliari, Italy; 3Scuola Specializzazione Medicina Interna, Universita’ di Cagliari, 09124 Cagliari, Italy; 4Dipartimento Scienze Mediche e Sanita’ Pubblica, Universita’ di Cagliari, 09124 Cagliari, Italy

**Keywords:** aging, vitamin B12, megaloblastic anemia, pancytopenia

## Abstract

Background: Vitamin B12 (cobalamin CBL) is a water-soluble vitamin required to form hematopoietic cells (red blood cells, white blood cells, and platelets). It is involved in the process of synthesizing DNA and myelin sheath. Deficiencies of vitamin B12 and/or folate can cause megaloblastic anemia (macrocytic anemia with other features due to impaired cell division). Pancytopenia is a less frequent exordium of severe vitamin B12 deficiency. Vitamin B12 deficiency can also cause neuropsychiatric findings. In addition to correcting the deficiency, an essential aspect of management is determining the underlying cause because the need for additional testing, the duration of therapy, and the route of administration may differ depending on the underlying cause. Methods: Here, we present a series of four patients hospitalized for megaloblastic anemia (MA) in pancytopenia. All patients diagnosed with MA were studied for a clinic-hematological and etiological profile. Results: All the patients presented with pancytopenia and megaloblastic anemia. Vitamin B12 deficiency was documented in 100% of cases. There was no correlation between the severity of anemia and deficiency of the vitamin. Overt clinical neuropathy was present in none of the cases of MA, while subclinical neuropathy was seen in one case. The etiology of vitamin B12 deficiency was pernicious anemia in two cases and low food intake in the remaining cases. Conclusion: This case study emphasizes the role of vitamin B12 deficiency as a leading cause of pancytopenia among adults.

## 1. Introduction

Malnutrition, often called insufficient nutrition, is a common condition with advancing age. Its prevalence in the elderly ranges from 5–10% in community-dwelling subjects, 30–60% in inpatients, and up to 85% in nursing home residents [1,2]. Malnutrition is associated with unfavorable outcomes in older subjects [3,4], post-operative delirium [5], and anemia [6].

The prevalence of anemia is also high in the elderly: from 17–24% in the community [7] to 55% in nursing homes [8]. Anemia increases the burden of any medical condition, including surgery [9], reduces the independence of older subjects [10], and is risky for depression and cognitive impairment [11]. In addition, anemia in older individuals can be secondary to potentially inappropriate prescribing [12], low protein intake and albumin levels [13], and selective nutritional deficiency [14].

Vitamin B12 deficiency increases with aging [15] and has been associated with megaloblastic anemia and/or overt neurological complications [16]. Anemia in vitamin B12 deficiency is characterized by ineffective erythropoiesis caused by intramedullary apoptosis of megaloblastic erythroid precursors [17] and/or hemolysis because of shortened red cell survival [18]; increased plasma bilirubin and serum lactic dehydrogenase (LDH), with usually normal AST levels [19]; and higher iron, ferritin, and soluble transferrin receptor as a feature of a block in iron utilization [20,21,22].

Below, we illustrate a series of four patients hospitalized over three months whose clinic presentation was characterized by vague symptoms and appeared as pancytopenia affecting contemporarily erythrocytes, leukocytes, and platelet. Our internal medicine ward is one of the largest ones in Cagliari and is the main reference point of care for a vast catchment area. Anemia is the third leading cause of hospitalization in our ward. One of the most common findings is high MCV and low B12 levels, but only the above-mentioned cases presented with pancytopenia.

Perhaps due to the pandemic, our patients neglected their periodic checkups and scheduled visits with the general practitioner. They underwent available blood exams after the lockdown, but it was too late.

## 2. Case Reports

### 2.1. Case 1

A 71-year-old Caucasian woman with multimorbidity presented with generalized fatigue and new onset dyspnea on exertion of three-week duration. She was hospitalized because of a hemoglobin level of 4.6 g/dL during a periodic blood test check.

On admission, both vital signs and physical examination were typical, maybe due to a long-standing condition. Further review of labs showed pancytopenia (Table 1). Vitamin B12 levels were low (75 pg/mL, normal range: 211–911), which were attributed to pernicious anemia (presence of intrinsic factor blocking and anti-parietal cell antibodies); lack of intrinsic factor and loss of parietal cells were highlighted from an endoscopic exam of the stomach. We performed the endoscopic procedure about five days after the ward admission as the hemoglobin level reached 8 gr/dL (the lower hemoglobin level acceptable for undergoing an endoscopic procedure safely). The histologic examination described the typical feature of atrophic gastritis, with atrophy of all surfaces, loss of gastric glands and parietal cells, and infiltration of the lamina propria by lymphocytes and plasma cells.

The patient began therapy with three units of EUC and intramuscular vitamin B12 supplementation every other day. Treatment benefitted all three hematological lines (Table 1 bottom lines), so the patient was discharged home with a weekly intramuscular vitamin B12 supplementation. After three months, a blood test showed values in the normal range: hemoglobin and hematocrit of 13.3 g/dL and 42.6%, respectively, with platelets of 357 × 10^3^ μL, MCV 82.7 fL, RBC 5.15 × 10^6^ μL, and WBC 8.19 × 10^3^ μL.

### 2.2. Case 2

An 81-year-old Caucasian woman with a past medical history of epilepsy presented at the medical ward with generalized fatigue and anorexia on exertion for six months. Her last lab results showed a level of hemoglobin of 3.7 g/dL, so she was hospitalized. On admission, her vitals and physical examination were normal, her BMI was 18, and her tongue was papillated at clinical examination. Her neurological exam did not suggest any neuropathy. Further review of labs showed pancytopenia (Table 1), with hemoglobin of 4.7 g/dL and a critical vitamin B12 deficiency at 94 pg/mL. She was tested for anti-intrinsic factor antibodies and anti-parietal gastric cells; the results for both were negative. The patient also performed an osteomidollar biopsy, which showed the presence of trilinear dysplasia, compatible with a severe cyanocobalamin deficiency. The biopsy also highlighted multi-nucleated and multi-segmented neutrophils. There was no observed hemosiderin deposition on red cell precursors. The M/E ratio was decreased.

The patient did not undergo endoscopic exams because her stool exam did not have blood, and we had a coherent explanation of her clinical condition. In this case, anorexia and low food intake were determined to be the causes of vitamin b12 deficiency. She also had severe vitamin D deficiency, but her iron deposit was high enough. Therefore, the patient began therapy with three units of EUC and intramuscular vitamin B12 supplementation every other day. With these treatments, her hemoglobin, RBC, WBC, and platelets improved, respectively, to 7.9 g/dL, 2.53 × 10^6^/microL, 4.47 × 10^3^/microL, and 154 × 10^3^/microL, so the patient was discharged home with a weekly intramuscular vitamin B12 supplementation. After three months, lab results showed values in the normal range: hemoglobin and hematocrit levels of 10.6 g/dL and 32.1%, respectively, with platelet levels of 193 × 10^3^ microL, MCV 100.9 fL, RBC 3.19 × 10^6^ microL, and WBC 3.50 × 10^3^ microL.

### 2.3. Case 3

A 64-year-old Caucasian male with spondylitis ankylosing presented with generalized fatigue and paresthesias in the lower limbs for two weeks. Blood test showed that hemoglobin was 4.6 g/dL. Upon hospital admission, his vitals and physical examination were normal. Further review of labs showed pancytopenia (Table 1) and low vitamin B12 levels (129 pg/mL, normal range: 211–911), attributable to pernicious anemia with anti-IF antibodies. After seven days of B12 supplementation (four units of cyanocobalamin 1000 UI and alternately daily intramuscular vitamin B12 supplementation), all three hematological lines showed significant improvement (Table 1, bottom lines), and the patient was discharged home. After three months of weekly intramuscular vitamin B12 supplementation, blood test results were within the normal range: hemoglobin and hematocrit of 14.5 g/dL and 43.5%, respectively, with MCV 83.8 fL, RBC 5.19 × 10^6^ microL platelets of 86 × 10^3^ microL, and WBC 6.45 × 10^3^ microL.

### 2.4. Case 4

An 84-year-old Caucasian male with multiple comorbidities presented with an accidental fall and hematuria. At hospital admission, hemoglobin of 3.3 g/dL was observed. On admission, his vitals and physical examination were normal. His BMI was 22. No signs of neuropathy were observable at the neurological examination. Further review of labs showed pancytopenia (Table 1) with vitamin B12 (55 pg/mL, normal range: 211–911) and folate (1.8 ng/mL, average value > 5.38) deficiency. Unfortunately, a blood smear was not available for this patient. The etiology was hyporexia with reduced food intake and then malnutrition. Vitamin B12 intramuscular supplementation yielded a relevant improvement in hemoglobin (10.4 g/dL), RBC (3.37 × 10^6^ microL), WBC (12.79 × 10^3^ microL), and platelets (175 × 10^3^/microL). Unfortunately, the patient’s relatives did not give us consent to execute an invasive endoscopic examination in light of the significant improvement in the clinical conditions following B12 supplementation.

## 3. Conclusions

Anemia is a common feature in older subjects, often leading to hospitalization. Vitamin B12 deficiency is recognized as a common cause of anemia, usually presenting with macrocytosis. Less frequent and less known is the presentation of B12 deficiency as pancytopenia (i.e., reduction in all three hematological lines without neurological manifestations). As recorded in two of the four patients, it is helpful to investigate the presence of autoantibodies against the intrinsic factor. The intrinsic factor is essential for the normal absorption of cyanocobalamin in the bowel tract. The presence of autoantibodies directed against the intrinsic factor and atrophic gastritis are the hallmarks of pernicious anemia. Autoimmune atrophic gastritis ranges from 0.1% to 1–2% in the general population; for women and people aged > 60, the prevalence is 2–3%, and the predominance is 2:1 for females over males [23].

Even in the presence of multimorbidity and/or severe hematological alterations, vitamin B12 supplementation quickly, effectively, and dramatically improves hemoglobin, WBC, and platelet levels.

## Figures and Tables

**Table 1 jcm-12-02059-t001:** Patients’ clinical features.

	Case 1	Case 2	Case3	Case 4
Age	71	81	64	84
Gender	female	female	male	male
Symptoms	Dyspnea, Asthenia	Asthenia, Anorexia	Fatigue/Paresthesias	Undeclared
Symptoms duration	three weeks	six months	two weeks	/
Hb at the ward admission (12–16.5 g/dL)	4.5	4.7	7.5	3.3
MCV (85–95 FI)	110	115	125	109
RBC (4.00–5.00 10^6^/µL)	1.36	0.98	1.87	0.95
WBC (4.00–10.00 10^3^/µL)	2.18	2.75	2.31	1.70
Neutrophilis (1.8–8.0 10^3^/µL)	1.2	/	/	/
Platelets (150–450 10^3^/µL)	90	55	28	70
Vitamin B12 (211–911 pg/mL)	75	94	129	55
Folate (>3.38 ng/mL)	23.4	7	/	1.8
Serum iron (50–170 mcg/dL)	182	90	273	101
Ferritin (10–291 ng/mL)	278	50	212	186
Iron saturation (250–380 mg/dL)	146	270	155	285
LDH (120–246 U/L)	2800	800	627	670
Haptoglobin (40–240 mg/dL)	<6	<6	<6	<6
Direct and indirect test di Coombs	negative	negative	negative	negative
Gastric parietal cells	positive	negative	negative	negative
Targeting intrinsic factor	negative	negative	positive	negative
Diagnosis	Pernicious anemia	Malnutrition	Pernicious anemia	Malnutrition
Hb at discharge (12–16.5 g/dL)	8	7.9	10.9	/
Hb after three months (12–16.5 g/dL)	13.3	10.6	14.5	/
RBC at discharge (4.00–5.00 10^6^/µL)	5.15	3.19	5.19	/
WBC at discharge (4.00–10.00 10^6^/µL)	5.15	3.50	6.45	/
Platelets at discharge (150–450 10^3^/µL)	357	193	86	/

## Data Availability

More data are available upon request to the corresponding author.

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
