# Peer review of "Pancytopenia Secondary to Vitamin B12 Deficiency in Older Subjects"

_jcm, 2023, doi:10.3390/jcm12052059_

Round 1

Reviewer 1 Report

In this manuscript, the authors present a series of 4 case reports. The cases include elderly patients with b12 deficiency and pancytopenia, 2 cases of pernicious anemia and 2 cases of nutritional b12 deficiency. The clinical details are very interesting and the data are generally clear and well presented. The English wording and style need editing and correction.

Criticisms and comments

1. introduction/conclusions:

a) The presentation of b12 deficiency as pancytopenia is rare. The authors should give some sense of the frequency of this presentation i.e. how many patients were seen in their clinical practice/hospital, how many cases of b12 deficiency were there, and how many cases of pancytopenia were there (presumably 4).

b) The remarkably asymptomatic nature of these patients should be highlighted. Presumably the lack of symptoms reflects the slow onset of symptoms and the physiological equilibration that occurred over time.

c) Some discussion of the nature of pernicious anemia should be included - i.e. this is an autoimmune disease targeting B12 absorption.

d) More discussion of the nutritional deficiency in cases 2 and 4 should be included. Did they have total total caloric intake deficiency, protein deficiency, vitamin intake deficiency, iron deficiency. Was the gut examined in cases 2 and 4 and were there abnormalities. 

2. request for additional clinical details.

For each of the 4 cases , the description indicates that vital signs and physical examination were normal. The vital signs should be shown. If the pulse is in fact normal with hgb 4.7, 3.7, 7.5, 3.3 g/dl, this fact in itself is remarkable. Nutritional state and appearance should be noted. In patients 2 and 4 with malnutrition, what was the weight, bmi, and overall appearance. Did they exhibit cachexia? Was the tongue papillated? On neurologic examination, the text states that overt clinical neuropathy was not present. What about more subtle signs of neuropathy - proprioception, position sense, reflexes, etc.

3. request for additional lab/imaging details

Blood smear appearance should be noted for all 4 patients if available.

Patient 1 apparently had an endoscopic examination that showed loss of parietal cells of the stomach. A detailed description of the procedure, appearance, and biopsy result should be included.

Patient 2 apparently had a bone marrow aspirate and biopsy ("osteomidollar biopsy"). The results should be described in detail. What was the cellularity, what was the ME ratio, was there megaloblastic change and in which lineages, was iron present in macrophages and/or in red cell precursors, etc.

4. chart with clinical information

a) Suggest that in the table of clinical data, the range for normal values should be shown in the left hand column.

b) For "Gastric parietal cells", this indicates antibody to Gastric parietal cells? If so what is considered to be a positive antibody titer?

For "Targeting intrinsic factor", this indicates antibody to intrinsic factor? If so what is considered to a positive antibody titer?

c) Please add LDH to the table. LDH is an indicator of ineffective erythropoiesis and is often extremely increased in pernicious anemia, distinguishing this entity from iron deficiency anemia

d) Please add serum iron, iron saturation, and ferritin levels. Iron deficiency is the most common cause of anemia, and iron levels should therefore be included here.

5. Mode of replenishment of b12.

a) In our clinical practice we often avoid giving b12 intramuscularly for the initial doses due to thrombocytopenia and worry about developing a hematoma. Could the authors comment on this point?

b) Case 3 was given "four units of Cianocoblamin 1000 UI and alternately daily intramuscular vitamin B12 supplementation.." Is there a difference between cyanocobalamin and the intramuscular B12 supplementation?

c) Was folic acid supplementation given? There is literature suggesting that replenishment of folic acid prior to B12 could precipitate a neurologic crisis. In these cases, we presume folic acid was given concurrently with the B12. Is this true?

d) In case 2, 3 months after b12 replenishment, hgb had not returned to normal, and was only 10.6 g/dl. Why was this? Was there another underlying cause for the anemia? 

Author Response

We modified the manuscript as detailed below, carefully taking into account the constructive and very useful comments provided.

POINT BY POINT REPLY

Thank you very much for your interesting comments and suggestions. Below we explain how we have incorporated them into this new version of the manuscript in order to improve the paper.

1. introduction/conclusions:

a) The presentation of b12 deficiency as pancytopenia is rare. The authors should give some sense of the frequency of this presentation i.e. how many patients were seen in their clinical practice/hospital, how many cases of b12 deficiency were there, and how many cases of pancytopenia were there (presumably 4).

We thank the reviewer for her/his comment, we agreed with this point; thus, we modified the text of the manuscript accordingly.

b) The remarkably asymptomatic nature of these patients should be highlighted. Presumably the lack of symptoms reflects the slow onset of symptoms and the physiological equilibration that occurred over time.

We thank the reviewer for her/his comment, we agreed with this point; thus, we modified the text of the manuscript accordingly

c) Some discussion of the nature of pernicious anemia should be included - i.e. this is an autoimmune disease targeting B12 absorption.

We thank the reviewer for her/his comment, we add more details as suggested.

d) More discussion of the nutritional deficiency in cases 2 and 4 should be included. Did they have total total caloric intake deficiency, protein deficiency, vitamin intake deficiency, iron deficiency. Was the gut examined in cases 2 and 4 and were there abnormalities.

We thank the reviewer for her/his comment, we agreed with this point; thus, we modified the text of the manuscript accordingly

2. request for additional clinical details.

For each of the 4 cases , the description indicates that vital signs and physical examination were normal. The vital signs should be shown. If the pulse is in fact normal with hgb 4.7, 3.7, 7.5, 3.3 g/dl, this fact in itself is remarkable. Nutritional state and appearance should be noted. In patients 2 and 4 with malnutrition, what was the weight, bmi, and overall appearance. Did they exhibit cachexia? Was the tongue papillated? On neurologic examination, the text states that overt clinical neuropathy was not present. What about more subtle signs of neuropathy - proprioception, position sense, reflexes, etc.

We thank the reviewer for her/his comment, we added the clinical details as requested.

3. request for additional lab/imaging details

Blood smear appearance should be noted for all 4 patients if available.

Patient 1 apparently had an endoscopic examination that showed loss of parietal cells of the stomach. A detailed description of the procedure, appearance, and biopsy result should be included.

Patient 2 apparently had a bone marrow aspirate and biopsy ("osteomidollar biopsy"). The results should be described in detail. What was the cellularity, what was the ME ratio, was there megaloblastic change and in which lineages, was iron present in macrophages and/or in red cell precursors, etc.

We thank the reviewer for her/his comment, we add more details as suggested

4. chart with clinical information

a) Suggest that in the table of clinical data, the range for normal values should be shown in the left hand column.

We thank the reviewer for her/ his suggestion. We amended the table accordingly.

b) For "Gastric parietal cells", this indicates antibody to Gastric parietal cells? If so what is considered to be a positive antibody titer?

Yes a positive antibody titer was considered > 1.5

For "Targeting intrinsic factor", this indicates antibody to intrinsic factor? If so what is considered to a positive antibody titer?

Yes a positive antibody titer was considered > 1.5

c) Please add LDH to the table. LDH is an indicator of ineffective erythropoiesis and is often extremely increased in pernicious anemia, distinguishing this entity from iron deficiency anemia

We thank the reviewer for her /his suggestion. We amended the table accordingly.

d) Please add serum iron, iron saturation, and ferritin levels. Iron deficiency is the most common cause of anemia, and iron levels should therefore be included here.

We thank the reviewer for her /his suggestion. We amended the table accordingly.

5. Mode of replenishment of b12.

The way of supplementation was for all patients intramuscular.

a) In our clinical practice we often avoid giving b12 intramuscularly for the initial doses due to thrombocytopenia and worry about developing a hematoma. Could the authors comment on this point?

We thank the reviewer for arising this point. We also concern about intramuscular supplementation but the available pharmacological form of cianocobalamin, in Italy, are only oral or intramuscular. Because of the poor absorption of cianocobalamin in our patient we prefer use the intramuscular way always with strong check of the administration point.

b) Case 3 was given "four units of Cianocoblamin 1000 UI and alternately daily intramuscular vitamin B12 supplementation.." Is there a difference between cyanocobalamin and the intramuscular B12 supplementation?

No it is the same.

c) Was folic acid supplementation given? There is literature suggesting that replenishment of folic acid prior to B12 could precipitate a neurologic crisis. In these cases, we presume folic acid was given concurrently with the B12. Is this true?

Yes we confirm that folic acid was given simultaneously to B12

d) In case 2, 3 months after b12 replenishment, hgb had not returned to normal, and was only 10.6 g/dl. Why was this? Was there another underlying cause for the anemia?

We thank the reviever for arising his/her interesting point. The lady is a etorozygote carrier of Beta Thalassemia so her normal Haemoglobin level is 10 – 10.5 gr/dl.

Hoping you will find the manuscript of interest for your distinguished Journal

Yours sincerely

Angelo Scuteri

Reviewer 2 Report

The main question addressed by the research is underdiagnosis of b12 deficiency in elder persons.

I consider the topic is relevant for patient care through better awareness.

It adds real life experience to the subject area compared with other published material.

The conclusions are consistent with the evidence and arguments presented.

However, it should be mentioned that upon diagnosis of pancytopenia leukemia is initially often suspected.

It should be mentioned that metformin (which is a very common dug used in treatment of diabetics) can also cause vitamin B12 deficiency (many doctors do not know this)

Author Response

We thank the Reviewer for her/his appreciation of our work.

Round 2

Reviewer 1 Report

In this manuscript the authors present 4 case reports, characterized by b12 deficiency and pancytopenia.  As noted, this is a rare presentation of b12 deficiency. In 2 cases, autoantibodies were present consistent with pernicious anemia,, and in 2 cases, severe malnutrition was the likely cause of the deficiency and pancytopenia. In all cases, a rapid and beneficial effect of vitamin b12 administration was observed. Queries from the prior review have been answered. The English language usage is not very good, and proof reading should be performed prior to publication. The case reports however are interesting and instructive for clinicians that care for patients with anemia. 

Author Response

We thank the Reviewer for her/his appreciation of our revision

We have extensively revised the English language